# Non-autoregressive Machine Translation with Probabilistic Context-free Grammar

**Shangtong Gui**[1,2,3]**,Chenze Shao**[2,3]**,Zhengrui Ma**[2,3]**,Xishan Zhang**[1,4]**,Yunji Chen**[1,3]**,Yang Feng**[2,3]*

[1]State Key Lab of Processors,
Institute of Computing Technology, Chinese Academy of Sciences
[2]Key Laboratory of Intelligent Information Processing,
Institute of Computing Technology, Chinese Academy of Sciences
[3]University of Chinese Academy of Sciences
[4]Cambricon Technologies

## Abstract

Non-autoregressive Transformer(NAT) significantly accelerates the inference of neural machine translation. However, conventional NAT models suffer from limited expression power and performance degradation compared to autoregressive (AT) models due to the assumption of conditional independence among target tokens. To address these limitations, we propose a novel approach called PCFG-NAT, which leverages a specially designed Probabilistic Context-Free Grammar (PCFG) to enhance the ability of NAT models to capture complex dependencies among output tokens. Experimental results on major machine translation benchmarks demonstrate that PCFG-NAT further narrows the gap in translation quality between NAT and AT models. Moreover, PCFG-NAT facilitates a deeper understanding of the generated sentences, addressing the lack of satisfactory explainability in neural machine translation.[2]

## 1 Introduction

Autoregressive machine translation models rely on sequential generation, leading to slow generation speeds and potential errors due to the cascading nature of the process. In contrast, non-autoregressive Transformer (NAT) models [13] have emerged as promising alternatives for machine translation. The Vanilla NAT model assumes that the target tokens are independent of each other given the source sentence. This assumption enables the parallel generation of all tokens. However, it neglects the contextual dependencies among target language words, resulting in a severe **multi-modality problem** [13] caused by the rich expressiveness and diversity inherent in natural languages. Consequently, Vanilla NAT models exhibit significant performance degradation compared to their autoregressive counterparts.

Several approaches have been proposed to alleviate the multi-modality problem in NAT models while maintaining their computational efficiency. CTC-based NAT [11, 22] depicted in Figure 1(b) addresses this issue by implicitly modeling linear adjacent dependencies through token repetition and blank probability. This approach helps alleviate the multi-modality of token distributions at each position compared to Vanilla NAT. Similarly, DA-Transformer [14, 37, 26] shown in Figure 1(c) introduces a hidden Markov model and employs a directed acyclic graph to model sentence probability, enabling the model to capture dependencies between adjacent tokens and reduce the multi-modal nature of the distribution.

---

*Corresponding author: Yang Feng

[2]Code is publicly available at `https://github.com/ictnlp/PCFG-NAT`.

37th Conference on Neural Information Processing Systems (NeurIPS 2023).

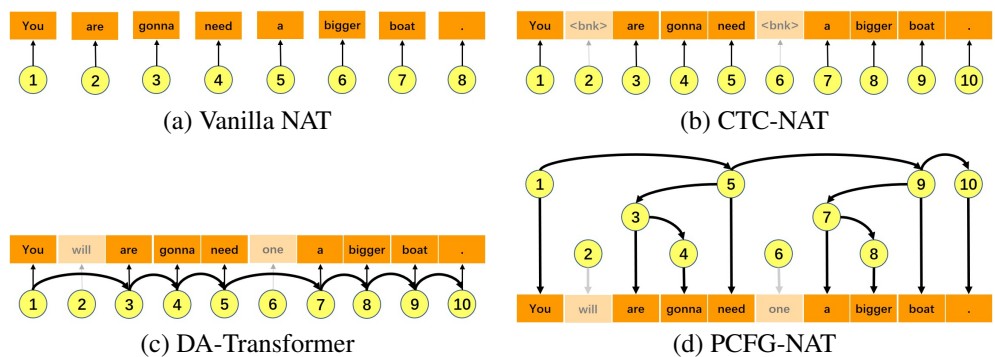

Figure 1: Illustration of Vanilla NAT, CTC-NAT, DA-Transformer, and PCFG-NAT

However, despite these enhancements, the modeling of only adjacent and unidirectional dependencies limits the model's ability to capture the rich semantic structures with non-adjacent and bidirectional dependencies present in natural language. For example, considering the token *need* as the previous token, the distribution of the next token is still affected by the issue of multi-modality, as the word *boat* can be modified by various rich prefixes in the corpus.

To overcome this constraints, we propose PCFG-NAT, a method that leverages context-free grammar to endow the model with the capability to capture the intricate syntactic and semantic structures present in natural language. For instance, as illustrated in Figure 1(d), PCFG-NAT can directly model the robust and deterministic correlation between non-adjacent tokens, such as *You need boat*. In the parse tree generated by PCFG-NAT for the target sentence, the nodes connected to the tokens *You need boat* constitute the main chain of the tree, capturing the left-to-right semantic dependencies through linear connections. Additionally, PCFG-NAT can learn relationships between modifiers, such as *are gonna* and *a bigger*, and the words they modify by constructing a local prefix tree for the nodes on the main chain. This allows for the modeling of right-to-left dependencies between the modified words and their modifiers.

We conduct experiments on major WMT benchmarks for NAT (WMT14 En↔De, WMT17 Zh↔En, WMT16 En↔Ro), which shows that our method substantially improves the translation performance and achieves comparable performance to autoregressive Transformer [40] with only one-iteration parallel decoding. Moreover, PCFG-NAT allows for the generation of sentences in a more interpretable manner, thus bridging the gap between performance and explainability in neural machine translation.

## 2 Background

In this section, we will present the necessary background information to facilitate readers' comprehension of the motivation and theoretical underpinnings of PCFG-NAT. In Section 2.1, we will provide a formalization of the machine translation task and introduce the concepts of autoregressive Transformer [40] and Vanilla NAT [13], which will serve as the basis for our proposed approach. Subsequently, in Section 2.2, we will provide a precise and rigorous definition of PCFG, accompanied by illustrative examples aimed at enhancing clarity and comprehension.

### 2.1 Non-autoregressive Translation

Machine translation can be formally defined as a sequence-to-sequence generation problem. Given source sentence $X = \{x_1, x_2, ..., x_S\}$, the translation model is required to generate target sentence $Y = \{y_1, y_2, ..., y_T\}$. The autoregressive translation model predicts each token on the condition of all previous tokens:

$$P(Y|X) = \prod_{t=1}^{T} p(y_t|y_{<t}, X). \tag{1}$$

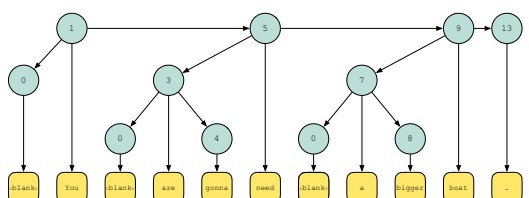

| Applied Rules | Sequence |
|---|---|
| Start | $V_1$ |
| $V_1 \rightarrow V_0$You $V_5$,$V_0 \rightarrow \epsilon$ | You $V_5$ |
| $V_5 \rightarrow V_3$ need $V_9$ | You $V_3$ need $V_9$ |
| $V_3 \rightarrow V_0$are $V_4$,$V_0 \rightarrow \epsilon$ | You are $V_4$ need $V_9$ |
| $V_4 \rightarrow$ gonna | You are gonna need $V_9$ |
| $V_9 \rightarrow V_7$ boat $V_{10}$ | You are gonna need $V_7$ boat $V_{10}$ |
| $V_7 \rightarrow V_0$a $V_8$,$V_0 \rightarrow \epsilon$ | You are gonna need a $V_8$ boat $V_{10}$ |
| $V_8 \rightarrow$ bigger | You are gonna need a bigger boat $V_{10}$ |
| $V_{10} \rightarrow$ . | You are gonna need a bigger boat . |

Figure 2: An example parse tree for **You are gonna need a bigger boat.**

Table 1: Derivation of the parse tree. Used production rules for every step are listed, where $\epsilon$ stands for an empty string.

The autoregressive inference is time-consuming as it requires generating target tokens from left to right. To reduce the decoding latency, Gu et al. [13] proposes non-autoregressive Translation, which discards dependency among target tokens and predicts all the tokens independently:

$$P(Y|X) = \prod_{t=1}^{T} p(y_t|X). \tag{2}$$

Since the prediction does not rely on previous translation history, the decoder inputs of NAT are purely positional embeddings or the copy of source embeddings. Vanilla NAT trains a length predictor to determine the sentence length when decoding. During the training, the golden target length is used. During the inference, the predictor predicts the target length and the translation of the given length is obtained by argmax decoding.

## 2.2 Probabilistic Context-free Grammar

Probabilistic Context-free Grammar is built upon Context-Free Grammar (CFG). A CFG is defined as a 4-tuple:

$$G = (\mathcal{N}, \mathcal{T}, \mathcal{R}, S),$$

where $\mathcal{N}$ is the set of non-terminal symbols, $\mathcal{T}$ is the set of terminal symbols, $\mathcal{R}$ is the set of production rules and $S$ is the start symbol. PCFG extends CFG by associating each production rule $r \in R$ with a probability $P(r)$. Production rules are in the form:

$$A \rightarrow \alpha, \tag{3}$$

where $A$ must be a single non-terminal symbol, and $\alpha$ is a string of terminals and/or nonterminals, $\alpha$ can also be empty. Starting with the sequence that only has start symbol $S$, we can replace any non-terminal symbol $A$ with $\alpha$ based on production rule $A \rightarrow \alpha$. The derivation process repeats until the sequence only contains terminal symbols and the derived hierarchical structure can be seen as a **Parse Tree**. Those production rules and their probabilities can be applied regardless of the contexts, which provides Markov property for the joint probability of the parse tree. For every rule, $A \rightarrow \alpha$ applied in a derivation, all of the symbols in $\alpha$ are the children nodes of $A$ in order. The probability of a parse tree $t$ is formulated as:

$$P(t) = \prod_{r \in R(t)} (P(r))^{n_r}, \tag{4}$$

where $R(t)$ is the set of production rules applied in $t$ and $n_r$ is the number of times that $r$ is applied. For sentence *You are gonna need a bigger boat.*, Figure 2 shows an example parse tree and how the sentence is derived from the start symbol $V_1$.

## 3 The Proposed Method

In the following sections, we present the formal definition of a novel variant of PCFG designed for PCFG-NAT in Section 3.1 and the architecture of the PCFG-NAT model in Section 3.2. Finally, we describe the training algorithm and decoding algorithm for PCFG-NAT in Section 3.3 and Section 3.4.

### 3.1 Right Heavy PCFG

Despite the strong expressive capabilities of PCFGs, the parameter induction process becomes challenging due to the vast latent variable space encompassing the possible parse tree structures of a single sentence. Moreover, training PCFGs incurs significant time complexity [35]. To address these challenges, we propose a novel variant of PCFG called Right-Heavy PCFG (RH-PCFG). RH-PCFG strikes a balance between expressive capabilities and computational complexity, mitigating the difficulties associated with parameter induction and reducing training time.

#### 3.1.1 Derived Parse Tree of RH-PCFG

RH-PCFG enforces a distinct right-heavy binary tree structure in the generated parse tree. Specifically, for each non-terminal symbol in the tree, the height of the right subtree increases proportionally with the length of the derived sub-string of the non-terminal symbol, while the height of the left subtree remains constant. This characteristic significantly reduces the number of possible parse trees for a given sentence, thereby reducing training complexity.

In the context of a right-heavy binary tree, the traversal process begins with the root node and proceeds recursively by selecting the right child of the current node as the subsequent node to visit. This sequential path, known as the **Main Chain**, is guided by a design principle inspired by linguistic observations in natural language. Specifically, it reflects the tendency for sentence structures to display left-to-right connectivity, with local complements predominantly positioned to the left of the central core structure. Within this framework, the left subtree of each node along the main chain is referred to as its **Local Prefix Tree**. The parse tree in Figure 2 represents a right heavy tree. In this tree, $V_1, V_5, V_9, V_{13}$ form the main chain, and their left subtrees correspond to their respective local prefix trees.

#### 3.1.2 Support Tree of RH-PCFG

To make sure that all the parse trees are the right heavy tree, we first build up a **Support Tree** as the backbone for the parse trees derived from RH-PCFG. The construction of the support tree follows two steps: At first, given a source sentence with a length of $L_x$ and an upsample ratio of $\lambda$, we sequentially connect $\lambda L_x + 1$ nodes with right links, which become the candidates for the main chain nodes in parse trees. Next, excluding the root node, we append a complete binary tree of depth $l$ as the left subtree to each node. For the root node, only one left child is added. The added nodes represent local prefix tree candidates to the attached main chain candidates. Consequently, for a source sentence of length $L_x$, the number of non-terminal symbols $m$ of the RH-PCFG is

$$m = \lambda L_x \cdot 2^l + 2. \tag{5}$$

The nodes within the support tree are assigned sequential indexes through an in-order traversal, and these indexes correspond to the corresponding non-terminal symbols sharing the same index. Detailed construction algorithm can be found in Appendix A.1. Figure 3 shows an example of PCFG parse tree with $L_x = 3, \lambda = 1, l = 2$.

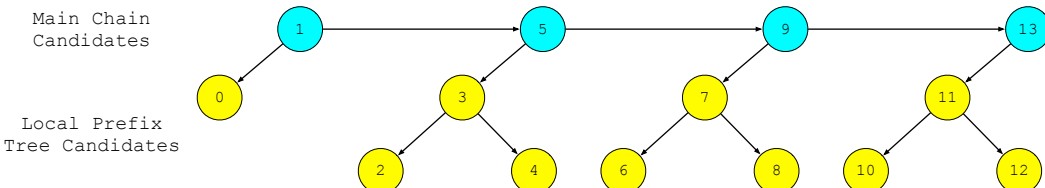

Figure 3: An example support tree of Right Heavy PCFG

In RH-PCFG, the set of non-terminals is $\mathcal{N} = \{V_0, ..., V_{m-1}\}$ and the set of terminals $\mathcal{T}$ is equivalent to the vocabulary. The collection of production rules is represented by $R$, and the starting symbol is $V_1$. The production rules in RH-PCFG have the following form:

$$V_0 \rightarrow \epsilon, \tag{6}$$

$$V_i \rightarrow a, a \in \mathcal{T}, i \neq 0, \texttt{IsLeaf}(i) \tag{7}$$

$$V_i \rightarrow V_j a V_k, V_i, V_j, V_k \in \mathcal{N}, a \in \mathcal{T}, (\texttt{InLeftReach}(j, i) \vee j = 0) \wedge \texttt{InRightReach}(k, i) \tag{8}$$

$\epsilon$ stands for an empty string. $\mathtt{IsLeaf}(i)$ indicates whether the non-terminal symbol $V_i$ corresponds to a leaf node in the support tree. $\mathtt{InLeftReach}(j, i)$ denotes that, in the support tree, the node corresponding to $V_j$ is in the left subtree of the node corresponding to $V_i$. $\mathtt{InRightReach}(k, i)$ signifies that the node corresponding to $V_k$ is in the right subtree of the node corresponding to $V_i$ in the support tree, and if $V_i$ corresponds to a main chain node, $V_k$ has to correspond to a main chain node at the same time.

## 3.2 Architecture

The main architecture of PCFG-NAT is identical to that of the Transformer [40], with the additional incorporation of structures to model the probabilities of the PCFG. The input sentence to be translated is fed into the encoder of the model. Additionally, based on the length $L_x$ of the source sentence, we compute the number of corresponding non-terminal symbols $m$ in the RH-PCFG. Then $m$ positional embeddings are used as inputs to the decoder. Through multiple layers of self-attention modules with a global attention view, cross-attention modules that focus on distant information, and subsequent feed-forward transformations, we obtain $m$ hidden states $h_0, ..., h_{m-1}$. These hidden states are associated with non-terminal symbols of the same index and are utilized to calculate the rule probabilities related to each corresponding non-terminal symbol.

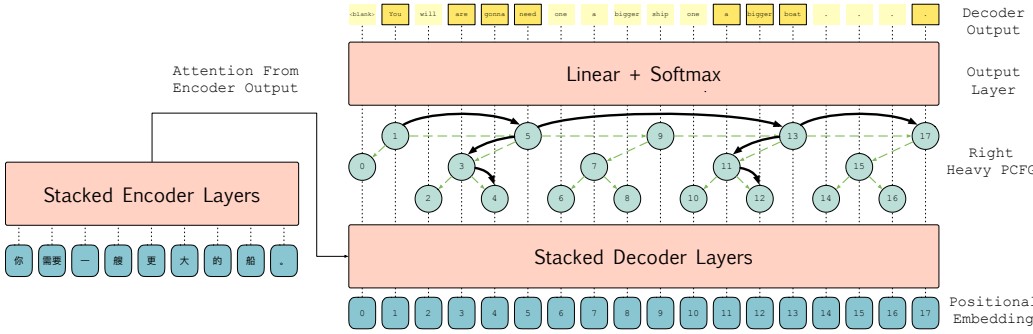

Figure 4: The architecture of PCFG-NAT model. A PCFG is inserted between the decoder layers and the output layer, which captures the dependency between different tokens. A parse tree is derived to generate the sentence *You are gonna need a bigger boat.* and is highlighted with bold black lines.

The probability of the rule $P(V_i \to a)$ can be regarded as the word prediction probability at each position, and we can directly calculate it with the softmax operation. We use $h_i$ to denote the hidden state representation of the non-terminal symbol $V_i$ and use $W_o$ to denote the weight matrix of the output layer. Formally, we have:

$$P(V_i \to a) = Softmax(W_o h_i). \tag{9}$$

For the rule $P(V_i \to V_j a V_k)$, we can independently consider the prediction probability of word $a$ and the transition probability between the three non-terminal symbols. Formally, we decompose it into two parts:

$$P(V_i \to V_j a V_k) = P(<V_j, V_k> | V_i) \cdot P(a | V_i). \tag{10}$$

The prediction probability $P(a | V_i)$ can be calculated with the softmax operation like in Equation 9. To normalize the distribution of $P(<V_j, V_k> | V_i)$, we have to enumerate all valid combinations of $j, k$. Let $\mathbf{Child}(i) = \{ <o, u> | V_i \to V_o a V_u \in \mathcal{R} \}$ denotes the set of children combinations of the non-terminal symbol $V_i$, we can formally define the transition probability as:

$$q_i = W_q h_i, \quad q_j = W_l h_j, \quad q_k = W_r h_k, \tag{11}$$

$$P(<V_j, V_k> | V_i) = \frac{\exp(q_i^T q_j + q_i^T q_k + q_j^T q_k)}{\sum_{<o,u> \in \mathbf{Child}(i)} \exp(q_i^T q_o + q_i^T q_u + q_o^T q_u)}, \tag{12}$$

where $W_q, W_l, W_r$ are learnable model parameters.

### 3.3 Training

#### 3.3.1 CYK Algorithm for RH-PCFG

We train PCFG-NAT with the maximum likelihood estimation, which sums up all the possible parse trees for the given sentence $Y$ conditioned on source sentence $X$ and model parameters $\theta$:

$$\mathcal{L}(\theta) = -\log P(Y|X, \theta) = -\log P(V_1 \Rightarrow y_0 y_1 ... y_{n-1}). \tag{13}$$

In PCFG, the CYK algorithm [35] is employed to compute the probability of a given sentence derived from the start symbol. The CYK algorithm follows a bottom-up dynamic programming approach to fill in the table $S$. Each element $S_{i,j}^a$ in the table represents the sum of probabilities of that $V_a$ derives the consecutive sub-string $y_i ... y_j$ of the target sentence, denoted as $P(V_a \Rightarrow y_i ... y_j)$. Therefore the size of table $S$ is $\mathcal{O}(mn^2)$. Based on the definition of $S_{i,j}^a$, we have recursion formula

$$S_{i,j}^a = \sum_{k=i+1}^{j-1} \sum_{<b,c> \in \mathbf{Child}(a)} P(V_a \to V_b y_k V_c) S_{i,k-1}^b S_{k+1,j}^c \tag{14}$$

For every non-terminal symbol $V_a$, the complexity filling all the $S_{i,j}^a$ is the product of the choice of span $< i, j >$, splitting point $k$, and children symbols $< b, c >$.

**Local Prefix Tree** For the non-terminal symbol $V_a$ in a local prefix tree, the rules specify that all values of $b$ and $c$ are strictly fewer than the size of the left attached tree in the support structure, denoted as $2^l - 1 < d = 2^l$. Similarly, the variables $i$, $j$, and $k$ are also strictly limited to values fewer than $d$, since it is the max length of sub-strings that $V_a$ can derives. Please note that $l$ represents the max depth of the complete binary tree added as the left subtree to each node. Therefore the complexity filling all the $S_{i,j}^a$ is $\mathcal{O}(d^5)$ for $V_a$ and all $V_a$ in Local Prefix Tree should be $\mathcal{O}(d^5 m)$ where $m$ is the number of non-terminal symbols.

**Main Chain** For the non-terminal symbol $V_a$ that serves as a main chain, based on the rules of the model, the variable $b$ has strictly fewer than $d$ choices, while $c$ has a maximum of $m/d$ choices. In a right-heavy parse tree, the symbols in the main chain are responsible for deriving a complete suffix of the target sentence as they are the right offspring of the root. Consequently, $j$ is always equal to $n - 1$, and variable $i$ has $n$ choices. However, $k$ has only $d$ choices since the the length of derived sub-string of $V_b$ is less than $d$. Considering the filling of all $S_{i,j}^a$ entries for the main chain symbols, the time complexity is $\mathcal{O}(mdn)$. Summing up the complexities for all $m/d$ main chain symbols, the total complexity is $\mathcal{O}(m^2 n)$.

Based on the above analysis, the overall time complexity of the CYK algorithm for RH-PCFG is $\mathcal{O}(d^5 n + m^2 n)$. Such complexity is within the acceptable range, while the training complexity of general form PCFG, such as Chomsky normal form [5], is $\mathcal{O}(n^3 |\mathcal{R}|)$. Detailed pseudocode for the training algorithm can be found in the Appendix A.2.

#### 3.3.2 Glancing Training

Glancing training, as introduced in the work of Qian et al. [30], has been successfully applied in various NAT architectures [12, 14]. In the case of PCFG-NAT, we also incorporate glancing training. This training approach involves replacing a portion of the decoder inputs with the embeddings of the target sentence $Y$. To determine the ratio of decoder inputs to be replaced with target embeddings, we employ the masking strategy proposed by Qian et al. [30]. There is a mismatch between the decoder length $m$ and target length $n$, so we have to derive an alignment between them to enable the glancing. To incorporate Glancing Training into PCFG-NAT, we adapt the approach by finding the parse tree $T_Y^* = \arg \max_{T_Y \in \Gamma(Y)} P(Y, T_Y | X)$ with the highest probability that generates the target sentence $Y$, where $\Gamma(Y)$ stands for all parse trees of $Y$. $T_Y^*$ can be traced by repalcing the **sum** opearation with **max** in the CYK algorithm. We provide the pseudocode in the Appendix A.3.

### 3.4 Inference

We propose a Viterbi [41] decoding framework for PCFG-NAT to find the optimal parse tree $T^* = \arg \max_T P(T|X, L)$ under the constraint of target length $L$. For each non-terminal symbol $V_a$, we

maintain a record $M_L^a$ that represents the highest probability among sub-strings of length $L$ that can be derived by $V_a$. Similar to the CYK algorithm, we can compute $M_L^a$ in a bottom-up manner using dynamic programming, considering all non-terminal symbols $V_a$ and lengths $L$.

$$M_L^a = \max_{<b,c>\in\mathbf{Child}(a),o\in\mathcal{T},k=1..L-1} P(V_a \to V_b o V_c) M_{L-1-k}^b M_k^c \tag{15}$$

To recover the parse tree with the maximum probability given $L$, we need to keep track of the values of $b$, $c$, and $k$ that lead to $M_L^a$ and start the construction from $M_L^1$. The detailed pseudocode is provided in the Appendix A.4. According to Shao et al. [37], we rerank the output sentences with probability and length factors and then output the sentence with the highest score.

## 4 Experiments

### 4.1 Setup

**Dataset** We conduct our experiments on WMT14 English↔German (En-De, 4.5M sentence pairs),WMT17 Chinese↔English (Zh-En, 20M sentence pairs) and WMT16 English↔Romanian (En-Ro, 610k sentence pairs). We use the same preprocessed data and train/dev/test splits as Kasai et al. [18]. For all the datasets, we learn a joint BPE model [36] with 32K merge operations to tokenize the data and share the vocabulary for source and target languages. The translation quality is evaluated with sacreBLEU [29] for WMT17 En-Zh and tokenized BLEU [28] for other benchmarks.

| Models | Iter | WMT14 | | WMT17 | | WMT16 | | Average Gap | Speedup |
|---|---|---|---|---|---|---|---|---|---|
| | | En-De | De-En | Eh-Zh | Zh-En | En-Ro | Ro-En | | |
| Transformer [40] | N | 27.66 | 31.59 | 34.89 | 23.89 | 34.26 | 33.87 | 0 | 1.0× |
| CMLM [8] | 10 | 24.61 | 29.40 | - | - | 32.86 | 32.87 | - | 2.2× |
| SMART [10] | 10 | 25.10 | 29.58 | - | - | 32.71 | 32.86 | - | 2.2× |
| DisCo [17] | ≈ 4 | 25.64 | - | - | - | - | 32.25 | - | 3.5× |
| Imputer [34] | 8 | 25.0 | - | - | - | - | - | - | 2.7× |
| CMLMC [15] | 10 | 26.40 | 30.92 | - | - | 34.14 | 34.13 | - | 1.7× |
| Vanilla NAT [13] | 1 | 11.79 | 16.27 | 18.92 | 8.69 | 19.93 | 24.71 | 14.31 | 15.3× |
| CTC [21] | 1 | 17.68 | 19.80 | 26.84 | 12.23 | | | - | 14.6× |
| FlowSeq[25] | 1 | 18.55 | 23.36 | - | - | 29.26 | 30.16 | - | 1.1× |
| AXE [9] | 1 | 20.40 | 24.90 | - | - | 30.47 | 31.42 | - | 14.2× |
| OAXE [6] | 1 | 22.4 | 26.8 | - | - | - | - | - | 14.2× |
| CTC + GLAT [30] | 1 | 25.02 | 29.14 | 30.65 | 19.92 | - | - | - | 14.6× |
| DA-Transformer + Lookahead [14] | 1 | 26.55 | 30.81 | 33.54 | 22.68 | 32.31* | 32.73* | 1.27 | 14.0× |
| DA-Transformer + Joint-Viterbi [37] | 1 | 26.89 | 31.10 | 33.65 | 23.24 | 32.46* | 32.84* | 1.00 | 13.2× |
| FA-DAT [26] | 1 | **27.47** | **31.44** | **34.49** | **24.22** | - | - | 0.41 | 13.2× |
| PCFG-NAT | 1 | 27.02 | 31.29 | 33.60 | 23.40 | **32.72** | **33.07** | 0.84 | 12.6× |

Table 2: Results on raw WMT14 En-De, WMT17 Zh-En and WMT16 En-Ro. 'Iter' means the number of decoding iterations, and $N$ is the length of the target sentence. The speedup is evaluated on the WMT14 En-De test set with a batch size of 1. '*' indicates the results of our re-implementation.

**Implementation Details** For our method, we choose the $l = 1, \lambda = 4$ settings for Support Tree of RH-PCFG and linearly anneal $\tau$ from 0.5 to 0.1 for the glancing training. For DA-Transformer [14], we use $\lambda = 8$ for the graph size, which has comparable hidden states to our models. All models are optimized with Adam [20] with $\beta = (0.9, 0.98)$ and $\epsilon = 10^{-8}$. For all models, each batch contains approximately 64K source words. All models are trained for 300K steps. For WMT14 En-De and WMT17 Zh-En, we measure validation BLEU for every epoch and average the 5 best checkpoints as the final model. For WMT16 En-Ro, we just use the best checkpoint on the valid dataset. We use the NVIDIA Tesla V100S-PCIE-32GB GPU to measure the translation latency on the WMT14 En-De test set with a batch size of 1. We implement our models based on the open-source framework of `fairseq` [27].

### 4.2 Main Results

**Translation Quality**

As demonstrated in Table 2, PCFG-NAT outperforms CTC-GLAT [30] and DA-Transformer [14] in terms of translation quality, effectively reducing the performance gap between NAT and AT models. This indicates its ability to capture the intricate translation distribution inherent in the raw training data and efficiently learn from dependencies with reduced multi-modality. FA-DAT [26] builds upon

| Decoding Strategy | WMT14 | | WMT17 | | WMT16 | | Speedup |
|---|---|---|---|---|---|---|---|
| | En-De | De-En | En-Zh | Zh-En | En-Ro | Ro-En | |
| Greedy Decoding | 26.01 | 31.38 | 32.82 | 22.50 | 32.05 | 32.65 | 1.00× |
| Viterbi Decoding | 27.02 | 32.29 | 33.60 | 23.40 | 32.72 | 33.07 | 0.91× |

Table 3: Ablation Study on the effect of decoding strategies. We use the NVIDIA Tesla V100S-PCIE-32GB GPU to measure the speedup on the WMT14 En-De test set with a batch size of 1.

DA-Transformer and trains the model to maximize a fuzzy alignment score between the graph and reference, taking captured translations in all modalities into account. The same training objective can also be implemented in PCFG-NAT. Experimental results show that FA-DAT [26] achieves state-of-the-art NAT performance. In the future, we plan to incorporate the ideas from FA-DAT [26] into PCFG-NAT, enabling a fairer comparison and demonstrating the superiority of PCFG modeling.

**Inference Latency**

As shown Table 2, PCFG-NAT demonstrates a decoding speedup that is slightly lower than Vanilla NAT but significantly higher than both AT models and iterative NAT models. This observation indicates that PCFG-NAT effectively strikes a favorable balance between translation quality and decoding latency, achieving a desirable trade-off between the two factors.

### 4.3 Ablation study

#### 4.3.1 Structure of Support Tree

In this section, we examine the impact of the max depth $l$ of the local prefix tree and the upsample ratio $\lambda$ in the support tree of RH-PCFG. Figure 5 presents the results. When the total number of non-terminal symbols is kept constant, incorporating a local prefix tree with max two layers leads to a significant decline in performance. We posit that an excessively large value of $l$ leads to a significant increase in the number of potential parse trees, thereby making it more challenging for the model to learn complex structures from the data. As a result, the translation performance of the model may decline.

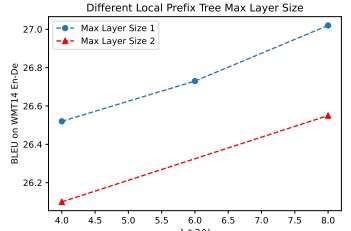

Figure 5: Ablation study on the max depth $l$ of local prefix tree and upsample rample ration $\lambda$. Notice the number of the non-terminal symbols $m$ equals to $\lambda L_x \cdot 2^l + 2$, where $L_x$ is the length of the source sentence.

### 4.4 Decoding Strategy

To determine the optimal parse tree for a given sentence length, we employ Viterbi decoding. However, this approach incurs a certain computational cost in terms of decoding speed. To demonstrate the necessity of Viterbi decoding, we compare it with a greedy decoding strategy that always selects the rule with the highest probability. The experimental results, presented in Table 3, indicate that while Viterbi decoding is only slightly slower than greedy decoding, it yields significantly better performance. Based on these findings, we assert that Viterbi decoding is necessary and do not recommend the use of greedy decoding.

### 4.5 Ablation Study for Distilled Data and GLAT

We conducted ablation studies to examine the impact of Glancing Training and Knowledge Distillation on our model's performance.GLAT consistently improves the translation performance of all models, while the gains from KD are more limited.

| WMT16 En-Ro | | BLEU |
|---|---|---|
| **Transformer [40]** | Baseline | 34.26 |
| | +KD | 34.72 |
| **DA Transformer [14]** | Baseline | 31.98 |
| | +GLAT | 32.46 |
| | +GLAT & KD | 32.40 |
| **PCFG-NAT** | Baseline | 31.94 |
| | +GLAT | 32.72 |
| | +GLAT & KD | 32.75 |

Table 4: Comparison of WMT16 En-Ro translation performance (BLEU scores) for Transformer, DA Transformer, and PCFG-NAT models with and without Glancing (GLAT) [30] and Knowledge Distillation (KD) techniques. Baseline means the models are trained on raw train sets without GLAT [30].

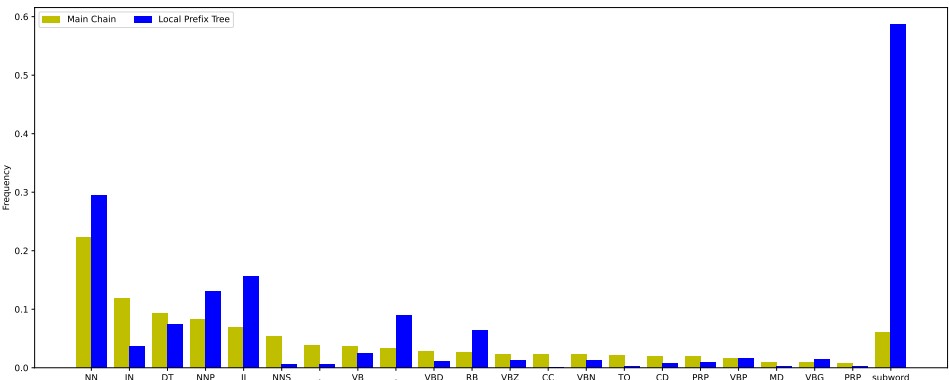

Figure 6: The frequency of part-of-speech and subwords generated from the main chain and local prefix tree respectively.

## 4.6 Syntax Analysis for Generated Parse Tree

To analyze the syntax information captured by PCFG-NAT, we conducted an analytical experiment on the WMT14 De-En test set. We utilized the averaged perceptron tagger, a part-of-speech tagging tool available in the nltk library [24], to assign part-of-speech tags to the translations. Subsequently, we calculated the frequency of part-of-speech tags for tokens directly generated from non-terminal symbols in both the main chain and the local prefix tree. Additionally, we counted the frequency of unfinished subwords ending with the symbol @@. To ensure comparability, the frequency scores were normalized by the number of tokens in the main chain and local prefix tree, respectively. The results are presented in Figure 6.

As depicted in Figure 6, the majority of tokens generated by non-terminal symbols in the local prefix tree correspond to the part-of-speech tags **RB** (adverb), **JJ** (adjective), **NNP** (proper noun, singular), and **comma**. Furthermore, the frequency of unfinished subwords in the local prefix tree symbols is notably higher compared to that in the main chain. It is well-known that **RB**, **JJ**, **NNP**, **comma**, and unfinished subwords are closely related to the local adjacent tokens. Hence, PCFG-NAT effectively captures the connection between these elements with the local prefix tree serving as complementary information to the main chain.

## 4.7 Probability Analysis of the Maximum Likelihood Path

We calculated the proportion of the maximum probability parse tree (path) that generates the references among all parse trees (paths) to investigate the benefits of the syntax tree learned by the RH-PCFG in comparison to the path in the Directed Acyclic Graph (DAG). We observed that the proportion of the highest probability parse tree for PCFG-NAT is generally higher than the proportion

| | WMT14 | | WMT17 | | WMT16 | | Average |
|---|---|---|---|---|---|---|---|
| | En-De | De-En | En-Zh | Zh-En | En-Ro | Ro-En | |
| DA-Transformer | 0.6026 | 0.7565 | 0.4899 | 0.5611 | 0.7948 | 0.7124 | 0.6529 |
| PCFG-NAT | 0.6421 | 0.7718 | 0.4681 | 0.5886 | 0.7764 | 0.7160 | 0.6605 |

Table 5: In the Valid Set, for the DA-Transformer model, we calculated $P(Y|X, A^*)/P(Y|X)$, where $X, Y$ represents the source text and the target text in the dataset, $A^* = argmax_A P(A|X, Y)$, and $A$ represents a hidden state path. For the PCFG-NAT model, we calculated $P(Y|X, T^*)/P(Y|X)$, where $T^* = argmax_T P(T|X, Y)$, and T represents a parse tree.

of the highest probability path for DA-Transformer [14]. This demonstrates that in the trained RH-PCFG, the probabilities of parse trees are more concentrated. Under the same training set and learnable parameters, this indicates that the modeling approach of RH-PCFG is more in line with the intrinsic structure of sentences compared to the flattened assumption of Directed Acyclic Graph.

## 5 Related Works

Gu et al. [13] accelerates neural machine translation with a non-autoregressive Transformer, which comes at the cost of translation quality. The major reason for the performance degradation is the multi-modality problem that there may exist multiple translations for the same source sentence. Many efforts have been devoted to enhancing the representation power of NAT. One thread of research introduces latent variables to directly model the uncertainty in the translation process, with techniques like vector quantization [16, 33, 3, 4], generative flow [25], and variational inference [38, 12]. Another thread of research explores some statistical information for intermediate prediction, including word fertility [13], word alignment [31, 39], and syntactic structures [2, 23]. Zhang et al. [43] explores the syntactic multi-modality problem in non-autoregressive machine translation The most relevant works to PCFG-NAT are CTC-based NAT [11, 22] and DA-Transformer [14, 37, 26], which enriches the representation power of NAT with a longer decoder. They can simultaneously consider multiple translation modalities by ignoring some tokens [11] or modeling transitions [14]. Fang et al. [7] proposes a non-autoregressive direct speech-to-speech translation model, which achieves both high-quality translations and fast decoding speeds by decomposing the generation process into two steps. Compared to previous NAT approaches, PCFG-NAT mitigates the issue of multi-modality in NAT by capturing structured non-adjacent semantic dependencies in the target language, leading to improved translation quality. Our work is also related to the line of work on incorporating structured semantic information into machine translation models[42, 19, 32, 1]. In contrast to these works, PCFG-NAT learns structured information unsupervised from data and can generate target tokens in parallel.

## 6 Conclusion

Non-autoregressive Transformer has higher inference speed but weaker expression power due to the lack of dependency modeling. In this paper, we propose PCFG-NAT to model target-side dependency by capturing the hierarchical structure of the sentence. Experimental results show that PCFG-NAT further enhances the translation performance of NAT models while providing a more interpretable approach for generating translations.

## 7 Limitations

Further research and exploration are needed to overcome the limitations encountered when attempting to induce complex PCFGs from data. Developing more advanced learning algorithms or exploring alternative approaches for modeling and capturing the rich syntactic structures of language may be necessary to unlock the full potential of PCFG-NAT and further enhance its translation capabilities.

# 8  Acknowledgement

This work is partially supported by the National Key R&D Program of China(under Grant 2021ZD0110102), the NSF of China(under Grants 61925208), CAS Project for Young Scientists in Basic Research(YSBR-029) and Xplore Prize. We thank all the anonymous reviewers for their insightful and valuable comments.

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

# A  Appendix

## A.1  Construction of Support Tree

`src_length` stands for the length of source sentence $L_x$.

`layer_size` stands for max depth of local prefix tree $l$.

`upsample_ratio` stands for hyperparameter $\lambda$.

```python
def build_support_tree(src_length, layer_size, upsample_lambda):
    main_chain_size = upsample_lambda * src_length + 1
    root = Node()
    root.leftChild = Node()
    now = root
    for i in range(main_chain_size):
        now.rightChild = Node()
        now = now.rightChild
        now.leftChild = full_binary_tree(layer_size)
    return root
```
Listing 1: Code for Constructing a Support Tree in Section 3.1.2

```python
def full_binary_tree(layer_size):
    if layer_size <= 0:
        return None
    root = Node()
    root.leftChild = full_binary_tree(layer_size-1)
    root.rightChild = full_binary_tree(layer_size-1)
    return root
```
Listing 2: Code for Constructing a Constructing a Full Binary Tree with deepth `layer_size`

The function `build_support_tree` returns the root node of the support tree.

## A.2  CYK Algorithm for RH-PCFG

$m, n$ stands for the numbers of non-terminal symbols and target sentence length.

`layer_size` stands for max depth of local prefix tree $l$.

`OpProbs[a][i]` represents $P(y_i|V_a)$.

`RuleProb[a][b][c]` represents $P(<V_b, V_c>|V_a)$.

`S[a][i][j]` represents $P(V_a \Rightarrow y_i, ..., y_j)$. `S` are initialized to 0.

```python
def local_prefix_tree_cyk(S, OpProbs, RuleProb, m, n, layer_size):
    for i in range(0,n):
        for a in range(0,m):
            S[a][i][i] = OpProbs[a][i]
    for span in range(2, 2**layer_size):
        for a in range(0,m):
            for i in range(0,n):
                for b,c in Child(a):
                    for k in range(i, i+span):
                        S[a][i][i+span] += RuleProb[a][b][c] \
                            * OpProbs[a][k] \
                            * S[b][i][k-1] * S[c][k+1][i+span]
```
Listing 3: Code for CYK Training of Local Prefix Tree in Section 3.3.1.

```python
def main_chain_cyk(S, OpProbs, RuleProb, m, n, layer_size):
    for a in range(0, m):
        S[a][n-1][n-1] = OpProbs[a][n-1]
    for start in range(n-2, -1, -1):
        for a in range(0, m):
```

```
6              for i in range(0,n):
7                  for b,c in Child(a):
8                      for k in range(start, start+2**layer_size):
9                          S[a][start][n-1] += RuleProb[a][b][c] \
10                              * OpProbs[a][k] \
11                              * S[b][start][k-1] * S[c][k+1][n-1]
```

Listing 4: Code for CYK Training of Main Chain in Section 3.3.1.

After applying `local_prefix_tree_cyk` and `main_chain_cyk`, `S[1][0][n-1]` contains the value of $P(V_1 \Rightarrow y_0, ..., y_{n-1})$

## A.3   Best Parse Tree for Glancing Training

$m, n$ stands for the numbers of non-terminal symbols and target sentence length.

`layer_size` stands for max depth of local prefix tree $l$.

`OpProbs[a][i]` represents $P(y_i|V_a)$.

`RuleProb[a][b][c]` represents $P(<V_b, V_c > |V_a)$.

`S[a][i][j]` represents $P(V_a \Rightarrow y_i, ..., y_j)$.

`Trace[a][i][j]`=True means that derivation $P(V_a \Rightarrow y_i, ..., y_j)$ is part of the maximum derivation of target sentence. `Trace` are initialized to `False`.

`A[k]`=a means that the token at position $k$ in target sentence should be aligned to non-terminal symbol $V_a$. A are initialized to 0.

```
1 def best_parse_tree(S, A, Trace, OpProbs, RuleProb, m, n, layer_size):
2      Trace[1][0][n-1] = True
3      for start in range(0, n-2):
4          for a in range(0, m):
5              if Trace[a][start][n-1]:
6                  max_b, max_c, max_k = argmax(
7                              RuleProb[a][b][c] \
8                              * OpProbs[a][k] \
9                              * S[b][start][k-1] * S[c][k+1][n-1] \
10                             for (b,c,k) in \
11                             zip((Child(a), \
12                             range(start, start+2**layer_size)))
13                         )
14                 Trace[max_b][start][max_k-1] = True
15                 Trace[max_c][max_k+1][n-1] = True
16                 A[max_k] = a
17
18     for span in range(2, 2**layer_size):
19         for a in range(0, m):
20             for i in range(0,n):
21                 if Trace[a][i][i+span]:
22                     max_b, max_c, max_k = argmax(
23                         RuleProb[a][b][c] \
24                         * OpProbs[a][k] \
25                         * S[b][i][k-1] * S[c][k+1][i+span] \
26                         for (b,c,k) \
27                         in zip(Child(a), range(i, i+span))
28                     )
29                 Trace[max_b][i][max_k-1] = True
30                 Trace[max_c][max_k+1][i+span] = True
31                 A[max_k] = a
```

Listing 5: Code for Glancing Training Best Parse Tree in Section 3.3.2.

After applying the algorithm, the tensor $A$ contains the best alignment of non-terminal symbols and target tokens.

### A.4 Algorithm for Inference

$m, n$ stands for the numbers of non-terminal symbols and target sentence length.

`layer_size` stands for max depth of local prefix tree $l$.

`S[a][i][j]` represents $P(V_a \Rightarrow y_i, ..., y_j)$.

`MaxOpProbs[a]` represents $maxP(y|V_a)$.

`MaxOpTokens[a]` represents $argmax_y P(y|V_a)$.

`RuleProb[a][b][c]` represents $P(<V_b, V_c>|V_a)$.

`MAX_P[a][L]` are the probability of $M_L^a$ of

$$M_L^a = \max_{<b,c>\in \textbf{Child}(a), o\in \mathcal{T}, k=1..L-1} P(V_a \rightarrow V_b o V_c) M_{L-1-k}^b M_k^c \tag{16}$$

and `MAX_K[a][L],MAX_B[a][L],MAX_C[a][L]` are corresponding value of $k, b, c$.

`output` is the target sentence generated by viterbi decoding algorithm.

```
def viterbi_decoding(S, MaxOpProbs, RuleProb, m, n, layer_size, MAX_P,
    MAX_K, MAX_B, MAX_C):
    for start in range(n-2, -1, -1):
        for a in range(0, m):
            max_value, max_b, max_c, max_k = argmax(
                RuleProb[a][b][c] \
                * MaxOpProbs[a] \
                * MAX_P[b][k-1] * MAX_P[c][k+1][n-1] \
                for (b,c,k) in \
                zip((Child(a), \
                range(start, start+2**layer_size)))
                )
            MAX_P[a][n-start] = max_value
            MAX_K[a][n-start] = max_k
            MAX_B[a][n-start] = max_b
            MAX_C[a][n-start] = max_c

    for span in range(2, 2**layer_size):
        for a in range(0, m):
            max_value, max_b, max_c, max_k = argmax(
                RuleProb[a][b][c] \
                * MaxOpProbs[a] \
                * MAX_P[b][k-1] * MAX_P[c][span-k] \
                for (b,c,k) \
                in zip(Child(a), range(i, i+span))
                )
            MAX_P[a][span] = max_value
            MAX_K[a][span] = max_k
            MAX_B[a][span] = max_b
            MAX_C[a][span] = max_c
```

Listing 6: Code for Viterbi Decoding Algorihtm in Section 3.4.

```
def parse_tree(a, output, prefix_length, length, MaxOpTokens, MAX_B,
    MAX_C, MAX_K):
    max_b, max_c, max_k = MAX_B[a][length], MAX_C[a][length], MAX_K[a
    ][length]
    output[prefix_length+max_k[length]] = MaxOpTokens[a]

    parse_tree(max_b, prefix_length, max_k-1)
    parse_tree(max_c, prefix_length+max_k, length-max_k)
```

First, we utilize the `viterbi_decoding` method to obtain the values of `MAX_B, MAX_C, MAX_K`. Subsequently, considering the length constraint denoted as $L$, we employ the `parse_tree` function with parameters set as follows: starting index of 1 as $V_1$, `output` to store the gener-

ated target sentence, `prefix_length` with an initial value of 0, a maximum limit of $L$ tokens, `MaxOpTokens,MAX_B,MAX_C,MAX_K`. This process effectively generates the desired target sentence.

## A.5 Case Study

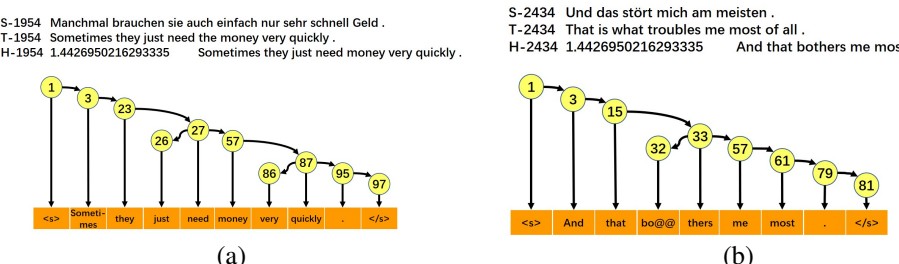

Figure 7: Two translation cases with generated parse tree from WMT14 De-En test set

To better understand the ability of PCFG-NAT to model structured semantic information, we select two cases from the WMT14 De-En test set and take a close look at the generated parse tree of PCFG-NAT. In Figure 7(a), we can see that in the sentence generated by PCFG-NAT, the content words lie in the main chain of the generated parse tree, and the words like **just, very**, which are supplements to words in the main chain, lie in the local prefix tree of the generated parse tree. In the generated parsing tree in Figure 7(b), the unfinished subword **bo@@** becomes the left node of its suffix **thers**, which demonstrates that PCFG-NAT can learn the local structure of natural language and enhances the interpretability of NMT to a certain extent.

