# OpenReview forum: "Non-autoregressive Machine Translation with Probabilistic Context-free Grammar"
_NeurIPS.cc/2023/Conference — NeurIPS 2023 poster_

### Official Review · Reviewer_8Jr4 · 2023-06-26

**Soundness:** 3 good
**Presentation:** 3 good
**Contribution:** 3 good
**Rating:** 6
**Confidence:** 4

**Summary:**

This work attempts to learn hierarchical structure in form of a PCFG as part of a non-autoregressive translation (NAT) system. The intuition is that the PCFG relaxes the conditional independence assumption between target tokens in vanilla NAT. The authors argue that related approaches with the same objective focus on local dependencies, whereas their hierarchical approach can also represent long range dependencies. Results seem to indicate a good balance between latency and translation quality.

**Strengths:**

The paper is well written and reproducible. The motivation for learning hierarchical dependencies in NAT is strong. The results are a bit on the slim side, but the novelty (as far as I am aware) of using hierarchy in NAT compensates for that. It is nice to see that a PCFG can be learned in that way. I appreciate the well designed ablation studies. The simplifying assumptions for the PCFG (binary and right-heavy) make sense.

**Weaknesses:**

I felt that the architecture under-delivers slightly on the claim of introducing PCFG-style dependencies to NAT, because production rule probabilities with a mix of non-terminals and terminals are still decomposed to the transition and terminal probabilities. The conditional independence assumption underlying Eq. 10 means that transition probabilities are independent of the terminal. Similarly, even if the PCFG introduces a link from y_i to y_j token, it doesn't mean that y_j really depends on y_i (because Eq. 9 is still used), just that the transition probability depends on h_i and h_j.

The connection to hierarchical SMT systems like Hiero is not made - please add citations.

**Questions:**

- Could you provide more details about the target length / number of non-terminal estimation in your system?
- Do you think there are any lessons learnt in hierarchical SMT that would be applicable here? For example, the constraints on production rules in Hiero?
- NAT and AT greedy/beam search lends itself particularly well to GPU/TPU decoding, whereas the Viterbi decoder seems harder to support efficiently on accelerators. Can you comment on that? Are the Table 2 speed-ups measured on the CPU, GPU, or TPU?

**Limitations:**

 Address last question above, if necessary.

---

> ### Author Rebuttal · Authors · 2023-08-10
>
> Thank you for your valuable feedback and comments on our paper. We appreciate your recognition of the novelty of our proposed PCFG-NAT method and its potential to learn hierarchical dependencies in NAT. We would like to address your concerns and questions in the following rebuttal.
>
> #### **PCFG-style dependencies in NAT:**
>
> We acknowledge that the decomposition of production rule probabilities with a mix of non-terminals and terminals into transition and terminal probabilities might not fully capture the PCFG-style dependencies in NAT. However, our method still provides a more flexible way to model dependencies between target tokens compared to vanilla NAT, as it allows the model to learn hierarchical structures and represent long-range dependencies. In future work, we will explore alternative approaches to better capture PCFG-style dependencies in NAT.
>
> #### **Connection to hierarchical SMT systems:**
>
> Thank you for pointing out the connection to hierarchical SMT systems like Hiero. Similar to the idea of Hiero, we control the complexity of parsing PCFG by fixing the number of non-terminal symbols in the production rules and the size of the production set. We also consider the alignment of terminal symbols to reduce the uncertainty of parse trees. We will include relevant citations in our revised paper and discuss the potential lessons learned from hierarchical SMT that could apply to our method, such as constraints on production rules.
>
> #### **Target length estimation and Viterbi decoder efficiency:**
>
> In our method, we use a upsample strategy to estimate the number of non-terminal symbols. $m = \lambda L_x \cdot 2^l + 2$.(Eq 5)
>
> During the decoding process, for every potential target length $L$, we use Viterbi decoding to find the optimal parse tree under the constraint of the target length $L$, as exhibited in Appendix A.4 of supplementary material. In accordance with Shao et al. [36], we reorder the output sentences of different lengths considering probability and length factors, ultimately outputting the sentence with the highest score.
>
> For the Viterbi decoder, we agree that it might be less efficient on accelerators like GPUs and TPUs compared to greedy/beam/Viterbi decoding. For now, we write a custom CUDA Kernal to complete the computation of Viterbi decoding, and it still has room for improvement. The speed-ups reported in Table 2 are measured on a GPU (NVIDIA Tesla V100S-PCIE-32GB). Despite the potential efficiency concerns, our Viterbi decoder still achieves a good balance between translation quality and decoding latency, as shown in our experimental results.
>
> We hope our responses have addressed your concerns and questions. We will incorporate these clarifications and improvements into the revised version of our paper. Thank you once again for your valuable feedback.

---

> > ### Comment · Reviewer_8Jr4 · 2023-08-21
> >
> > I've read the rebuttal and keep my score

---

> > > ### Author Response · Authors · 2023-08-21
> > >
> > > Thank you for taking the time to review our rebuttal. We appreciate your valuable feedback and will continue to refine our paper based on your suggestions.

---

### Official Review · Reviewer_sYwF · 2023-07-04

**Soundness:** 3 good
**Presentation:** 3 good
**Contribution:** 2 fair
**Rating:** 5
**Confidence:** 4

**Summary:**

This paper proposes PCFG-NAT, which alleviates the issue of conditional independent modeling through probabilistic context-free grammar. PCFG-NAT leverages syntax information to capture bidirectional dependencies among predicted tokens. In this way, this method also provides a better explanation of the translation process. Experiments on several benchmarks demonstrate the effectiveness of the proposed method in bridging the performance gap with autoregressive counterparts.

**Strengths:**

This paper proposes a novel PCFG-NAT method that introduces context-free grammar to capture dependencies in NAT. The authors develop systematic designs for it, including a lightweight variant called RH-PCFG, architecture support, training strategy, and decoding algorithms.

In addition, popular NAT techniques, such as glancing training and Viterbi decoding, can be integrated into PCFG-NAT.


**Weaknesses:**

The main concern for me is the wrong comparison. Table 2 shows the main results of the proposed method and related work. However, I notice that the authors omit some of the earlier methods that achieve better performance, such as [1][[2].
In addition, some reported results are inconsistent with the original paper, like the CTC+GLAT method reported in the paper [3] achieves 26.39 and 29.54 on WMT 14 En-De and De-En datasets, but 25.02 and 29.14 in this paper. If the report is reasonable, the author must explain it for clarification.

I admit that the proposed method is novel, but it is complex. Simple methods, like glancing training [3] and DSLP [2], achieve comparable performance without complex modifications.
According to Figure 2, PCFG is used only before the final prediction. I suspect that this design only helps slightly because the decoder layers are unchanged. Due to the complex design, it is difficult to design the ablation study to examine the role of PCFG.


[1] Fully non-autoregressive neural machine translation: tricks of the trade.

[2] Non-Autoregressive Translation with Layer-Wise Prediction and Deep Supervision.

[3] Glancing Transformer for Non-Autoregressive Neural Machine Translation.


**Questions:**

Can the authors elaborate the effect of each technique by detailed ablation studies? I am curious about the performance and speedup without glancing training or RH-PCFG.

---

> ### Author Rebuttal · Authors · 2023-08-10
>
> We appreciate your valuable feedback on our paper and would like to respond to your comments and suggestions.
>
> #### **Comparison with earlier methods and reported results:**
>
> Regarding the discrepancy in the reported results for the CTC+GLAT method, we would like to clarify that the difference is due to the fact that the original paper [3] used a knowledge-distilled training set, while our experiments were conducted using the undistilled training set. If this misunderstanding was your primary concern, we hope that you can reconsider the evaluation of our work after our clarification.
>
> While we agree that simple and effective methods are of great value, we believe that the complexity of our proposed PCFG-NAT method also has its merits. Our method explores a new probabilistic model for capturing natural language sequences. Although the performance improvement is limited, PCFG-NAT learns how to generate sentences in a structured and interpretable manner without supervision, which could provide valuable insights for future work. Indeed, the PCFG is only utilized before the final prediction and the decoder layers remain unchanged, but our main contribution lies in designing a new and efficient probabilistic model for capturing target-side dependencies in NAT. In future work, we will consider investigating more suitable decoder layer architectures to further enhance the performance of our model.
>
>
> #### **Ablation studies to examine the effect of each technique:**
>
> We acknowledge the importance of conducting detailed ablation studies to understand the impact of each technique, such as glancing training and RH-PCFG, on the performance and speedup of our method. Incorporating glancing training into PCFG-NAT encourages the model to capture more accurate dependencies between target tokens, resulting in improved translation quality compared to training without glancing, as shown in our global rebuttal. We will include additional experiments on larger datasets in the final version of our paper.
>
>
> We hope our responses have addressed your concerns and questions. We will incorporate these clarifications and improvements into the revised version of our paper. Thank you once again for your valuable feedback.

---

> > ### Comment · Reviewer_sYwF · 2023-08-16
> > **I Have Increased the Score**
> >
> > Some of my concerns have been addressed, and I'd like to increase my score. But problems remain after the rebuttal, as it stands now. It is a borderline accept submission in its current form.

---

> > > ### Author Response · Authors · 2023-08-21
> > >
> > > Thank you for taking the time to review our rebuttal and for considering an increase in your score. We appreciate your valuable feedback and acknowledge that there may still be some concerns after our rebuttal. We are committed to addressing these remaining issues and refining our paper to ensure it meets the highest standards. Your insights have been instrumental in helping us improve our work.

---

### Official Review · Reviewer_8FYH · 2023-07-05

**Soundness:** 3 good
**Presentation:** 2 fair
**Contribution:** 3 good
**Rating:** 6
**Confidence:** 3

**Summary:**

This paper proposes to rely on PCFG to incorporate the most likely parsing tree into the non-autoregressive transformer model as a way to mitigate multi-modality problem. Since considering all possible parsing trees is computationally expensive, the authors propose a right-heavy PCFG based on the assumption that language tends to show left-to-right connectivity. The experiments on the three well-known WMT datasets show their method outperforms others while keeping the speed-up provided by NAT.

**Strengths:**

- Novel approach to capture dependencies between tokens for NAT using PCFG
- Experimental results show that method allows to improve NAT translation in terms of BLEU score

**Weaknesses:**

- Limited analysis:
  - method is based on the assumption that left-to-right connectivity is the best for language and that "RH-PCFG strikes a balance between expressive capabilities and computational complexity"; however, no experiments support this claim.
  - Also, Figure 5 shows that increasing the local prefix tree helps in terms of BLEU.
- Missing references and comparison:
  - Follow-up work on DAG Ma, Zhengrui et al. "Fuzzy Alignments in Directed Acyclic Graph for Non-Autoregressive Machine Translation." (ICLR 2023, published Feb 1st). I understand this work appeared a few months before your submission, but it would be fair to compare.
- Lack of clarity
  - I have a hard time understanding the central concept of training and evaluating NAT model, which hurts reproducibility (please see questions)

**Questions:**

- What is your full objective function? During training, you rely on the target length, but you need to obtain the length from the model during decoding. Also, it is not clear to me if you only use the sum of LL of all possible trees as your only objective to train the translation model.
- Why do you use full vocabulary and not `<eos>` symbol as terminal states?
- Figure 5: it's unclear what contributes more, $\lambda$ or $l$. And in your plot, there is no plateau, meaning it could be better if you increase it further.
- I assume you do not use KD for your experiments. However, it is not explicitly stated. That is a valuable result that can strengthen your paper. Also, since KD is widely used, it would be interesting to see if KD can further improve your model

Typos & Grammar & Style:
- line 243: Figure instead of Table
- line 23: to mitigate.... to alleviate: duplication of the same meaning
- line 97: `know as` - I am confused here, I assume you propose this method, so it's not known before
- line 294: space after `quality.`

**Limitations:**

- Assumption that left-to-right dependency is more valuable for languages (which potentially false for some languages) is not addressed

---

> ### Author Rebuttal · Authors · 2023-08-10
>
> We appreciate your valuable feedback on our paper and would like to respond to your comments and suggestions.
>
> #### **Balancing between expressive capabilities and computational complexity:**
>
> We acknowledge that our paper could have provided more experimental evidence to support our claim that RH-PCFG strikes a balance between expressive capabilities and computational complexity. RH-PCFG maintains the expressive power of PCFG while reducing the training complexity. From the perspective of RH-PCFG improving translation performance, it can be understood that the expressive capacity of RH-PCFG enhances the translation quality of NAT, while the training complexity of RH-PCFG is lower than that of the general type of PCFG without rule pruning.
>
> #### **Questions about Figure 5:**
>
> Given that $m = \lambda L_x \cdot 2^l + 2$, under the condition of constant $l$, increasing $\lambda$ does enhance the BLEU score. Subsequently, we will continue to conduct experiments to investigate the impact of larger $\lambda$ values on the results. However, when the total number of non-terminal symbols $m$ remains constant, elevating $l$ results in considerable deterioration in performance. We hypothesize that an excessively large value of $l$ induces a significant increment in the number of potential parse trees, thereby posing increased difficulty for the model to discern complex structures from the data. Consequently, the translation performance of the model may experience a decline.
>
> #### **Missing references and comparison:**
>
> Thank you for pointing out the relevant work by Ma, Zhengrui et al. (ICLR 2023). We will include this reference in our revised paper and conduct a comparison with their work to provide a more comprehensive evaluation of our method.
>
> #### **Lack of clarity and objective function:**
>
> We apologize for any confusion regarding our objective function. During training, we maximize the likelihood of the target sentence by summing up all the possible parse trees for the given sentence conditioned on the source sentence and model parameters (Equation 6 in the paper) and the pseudocode is provided in supplementary material's Appendix A.2. During the decoding process, for every potential target length $L$, we use Viterbi decoding to find the optimal parse tree under the constraint of the target length $L$, as exhibited in Appendix A.4. Following Shao et al. [36], we reorder the generated sentences with different lengths considering probability and length factors, ultimately outputting the sentence with the highest score.
>
> Regarding the use of the full vocabulary and not the <eos> symbol as terminal states, terminal states are the symbols that will not derive a subtree. Consequently, for any PCFG responsible for generating sentences, the terminal states encompass the entirety of the vocabulary of the language.
>
> #### **Knowledge distillation:**
>
> In our experiments, we did not use KD for training our PCFG-NAT model. We agree that it is a valuable result that our method achieves competitive performance without relying on KD, which is widely used in other NAT models. This finding highlights the effectiveness of our proposed method in capturing hierarchical dependencies and mitigating the multi-modality problem in NAT.
>
> In our global rebuttal, we acknowledge that the effect of KD on performance is limited on the WMT16 EnRo dataset. We will include additional experiments on larger datasets in the final version of our paper.
>
> In the revised version of our paper, we will clarify the non-use of KD in our experiments and discuss the benefits of incorporating KD.
>
> #### **Typos, grammar, and style:**
>
> We appreciate your careful review and identification of these issues. We will correct the typos, grammar, and style issues in the revised version of our paper.
>
> #### **Limitations:**
>
> We acknowledge the limitation regarding our assumption that left-to-right dependency is more valuable for languages. To the best of our knowledge, left-to-right languages account for the largest proportion of the world's population, and the languages used in our experiments with the WMT Benchmark are also left-to-right. For the languages that conform to the right-to-left assumption, our method can be easily adapted to a right-to-left mode. In the future, we will conduct additional experiments to demonstrate this adaptability for a wider range of languages.
>
> We hope our responses have addressed your concerns and questions. We will incorporate these clarifications and improvements into the revised version of our paper. Thank you once again for your valuable feedback.

---

> > ### Comment · Reviewer_8FYH · 2023-08-14
> > **Changing my overall score to 6**
> >
> > Thank you for your reply and providing additional results.
> >
> > I am willing to increase my score to 6.

---

> > > ### Author Response · Authors · 2023-08-21
> > >
> > > Thank you for taking the time to review our rebuttal and for considering our clarifications and additional results. We appreciate your willingness to increase your score to 6. Your valuable feedback has helped us improve our work, and we will continue to refine our paper based on your suggestions.

---

### Official Review · Reviewer_EKPB · 2023-07-07

**Soundness:** 3 good
**Presentation:** 3 good
**Contribution:** 3 good
**Rating:** 6
**Confidence:** 5

**Summary:**

The paper proposes a new approach called PCFG-NAT to address the limitations of conventional Non-autoregressive Transformer (NAT) models. PCFG-NAT uses a specially designed Probabilistic Context-Free Grammar (PCFG) to capture complex dependencies among output tokens and enhance the expression power of NAT models. Experimental results on major machine translation benchmarks demonstrate that PCFG-NAT further narrows the gap in translation quality between NAT and autoregressive (AT) models. Additionally, PCFG-NAT facilitates a deeper understanding of the generated sentences and addresses the lack of satisfactory explainability in neural machine translation.

**Strengths:**

The paper proposes a very novel NAT model that performs comparably or even better than state-of-the-art NAT models.

**Weaknesses:**

1. The analysis of training complexity is not thorough enough.
a. It does not show the training inference of the proposed model compared with baselines.
b. Although the paper mentions that RH-PCFG is faster than PCFG, there is no quantitative analysis and no empirical evidence to support this claim. It would be helpful to compare the training complexity with the baseline and provide a more in-depth analysis.
c. How is d set in experiments? what is the relationship between d and m (or n)?

2. There is a lack of an in-depth explanation of the generated sentences from the main chain and prefix tree in the syntax analysis experiment. Only the analysis based on the frequency is not intuitive.  Additionally, the analysis of the syntax tree of the generated sentences in the experiment is not adequate, because some of such sentences suffer from grammar issues. It would be better to analyze the syntax tree of the reference translation based on the given <X, R> and calculate p(T|X,R).

3. Although RH-PCFG has strong expression power, the performance improvement is not significant. It would be helpful to explain the main reasons for this and whether there is potential to further explore this framework.

**Questions:**

N/A

**Limitations:**

See weakness

---

> ### Author Rebuttal · Authors · 2023-08-10
>
> We appreciate your valuable feedback on our paper and would like to respond to your comments and suggestions.
>
> #### **Training complexity analysis:**
>
> In our paper, we provided the algorithm and complexity analysis for both training and inference of PCFG-NAT, along with detailed pseudocode in the supplementary material's appendix. We would like to clarify that the training complexity of our PCFG-NAT model is $O(d^5n+m^2n)$, where $m$ is the number of non-terminal symbols, $n$ is the sentence length, and $d=2^l$, where $l$ is the maximum height of the local prefix tree. In comparison, the complexity of a PCFG without rule pruning of Right Heavy Support Tree is $O(n^3m^3)$. And the training complexities of CTC-NAT and DA-Transformer are $O(mn)$ and $O(m^2n)$ respectively. In future work, we plan to provide quantitative analysis to support our claim that RH-PCFG is faster than PCFG without rule pruning and compare RH-PCFG with other baselines.
>
> #### **Analysis of the syntax tree:**
>
> We agree that our syntax analysis experiment could be more comprehensive and intuitive. Analyzing the syntax tree of the reference translation based on the given <X, R> and calculating p(T|X,R) will provide a more in-depth understanding of the generated sentences from the main chain and prefix tree.
> In our global rebuttal, we have added a new set of experiments that show, in comparison to the non-hierarchical DAG model, RH-PCFG learns more concentrated probabilities for sentence structures. This indirectly demonstrates that RH-PCFG possesses a stronger ability to model dependencies between tokens. In the updated version of our manuscript, we will undertake in-depth experiments to analyze this aspect.
>
>
> #### **Performance improvement:**
>
> While our PCFG-NAT model demonstrates a performance improvement compared to other NAT models, we agree that there is still potential for further exploration and enhancement. The limited performance improvement could be due to the simplifications made in the RH-PCFG structure and the trade-offs between model complexity and performance. In future work, we plan to investigate alternative approaches to capture more complex dependencies and further improve the performance of our model.
>
> We appreciate your feedback and will incorporate these clarifications and improvements into the revised version of our paper.

---

> > ### Comment · Reviewer_EKPB · 2023-08-21
> > **Thanks for the rebuttal**
> >
> > I read the rebuttal and I keep my recommendation score.

---

> > > ### Author Response · Authors · 2023-08-21
> > >
> > > Thank you for taking the time to review our rebuttal. We appreciate your valuable feedback and will continue to refine our paper based on your suggestions.

---

### Official Review · Reviewer_FGhz · 2023-07-07

**Soundness:** 2 fair
**Presentation:** 3 good
**Contribution:** 2 fair
**Rating:** 5
**Confidence:** 4

**Summary:**

This article proposes a new NAT training method based on PCFG. To address the limitations of critical and unidirectional dependencies, the authors provide a new way of establishing dependencies from the perspective of CFG, hoping to capture both grammatical and semantic information. Therefore, this article simplifies PCFG into Right Heavy PCFG and, in the process of training and decoding generation in NAT, achieves a further reduction in the translation gap between NAT and autoregressive models by adding a derivation process based on Right Heavy PCFG.

**Strengths:**

An interesting algorithm based on PCFG is proposed, which reduces many limitations on previous methods and provides more semantic and grammatical dependencies.

**Weaknesses:**

1.	This paper claims to endow the capability in capture more complex dependencies, but the experimental part does not give a comparison and analysis of this aspect.
2.	The training loss function is not clear.


**Questions:**

1.	How does glancing training strategies influence the PCFG-NAT training?
2.	Which languages does the Right Heavy PCFG proposed in this paper suitable for?

---

> ### Author Rebuttal · Authors · 2023-08-10
>
> We appreciate your constructive feedback on our paper and would like to respond to your comments and concerns.
>
> #### **Capturing complex dependencies:**
> We acknowledge that our experimental section could be more explicit in comparing and analyzing the ability of our proposed PCFG-NAT method to capture complex dependencies. In the updated version of our manuscript, we will conduct in-depth experiments to analyze this aspect.
>
> Our syntax analysis experiment shows that our model generates more adverbs, adjectives, proper nouns, commas, and unfinished words from the local prefix tree. The semantic differences between the tokens generated by the main chain and local prefix tree indicate the ability of PCFG-NAT models to capture complex dependencies.
>
> In our global rebuttal, we have added a new set of experiments that demonstrate, in comparison to the non-hierarchical DAG model, RH-PCFG learns more concentrated probabilities for sentence structures. This indirectly shows that RH-PCFG possesses a stronger ability to model dependencies between tokens.
>
> #### **Glancing training strategies:**
>
> Incorporating glancing training into PCFG-NAT encourages the model to capture more accurate dependencies between target tokens, resulting in improved translation quality compared to training without glancing, as shown in the table in the global rebuttal. We will include additional experiments on larger datasets in the final version of our paper.
>
> #### **Suitability of Right Heavy PCFG for different languages:**
>
> The design of RH-PCFG is based on the assumption of left-to-right semantics in natural languages. To the best of our knowledge, left-to-right languages account for the largest proportion of the world's population, and the languages used in our experiments with the WMT Benchmark are also left-to-right. We acknowledge that there are languages that do not conform to the left-to-right assumption. However, our method can be easily adapted to other modes like right-to-left. In the future, we will conduct additional experiments to demonstrate this adaptability for a wider range of languages in the context of non-autoregressive translation.
>
> We appreciate your feedback and will incorporate these clarifications and improvements into the revised version of our paper.

---

### Author Rebuttal · Authors · 2023-08-10

In response to the reviewers' suggestions, we have incorporated two sets of analysis experiments during the rebuttal stage.

First, we conducted ablation studies to examine the impact of Glancing Training and Knowledge Distillation on our model's performance.GLAT consistently improves the translation performance of all models, while the gains from KD are more limited.

Secondly, we calculated the proportion of the maximum probability parse tree (path) that generates the references among all parse trees (paths) to investigate the benefits of the syntax tree learned by the RH-PCFG in comparison to the path in the Directed Acyclic Graph (DAG). We observed that the proportion of the highest probability parse tree for PCFG-NAT is generally higher than the proportion of the highest probability path for DA-Transformer. This demonstrates that in the trained RH-PCFG, the probabilities of parse trees are more concentrated. Under the same training set and learnable parameters, this indicates that the modeling approach of RH-PCFG is more in line with the intrinsic structure of sentences compared to the flattened assumption of Directed Acyclic Graph.

---

### Decision · Program_Chairs · 2023-09-21

**Decision:**

Accept (poster)

**Comment:**

The paper proposes a new approach called PCFG-NAT to address the limitations of conventional Non-autoregressive Transformer (NAT) models. Experimental results on major machine translation benchmarks demonstrate that PCFG-NAT further narrows the gap in translation quality between NAT and autoregressive (AT) models.

In all, the method is novel and reasonable. The main concerns are the empirical analyses. The author's response addressed the reviewers' concerns.